# A Simulation Study of the Dynamical Control of Optical Skyrmion Lattices through the Superposition of Optical Vortex Beams

Gao Tang [1], Chunyan Bai [2], Tianchen Tang [1], Jiansheng Peng [3], Songlin Zhuang [1,4] and Dawei Zhang [1,4,*]

1 School of Optical-Electrical and Computer Engineering, University of Shanghai for Science and Technology, Shanghai 200093, China; 222180386@st.usst.edu.cn (G.T.); 211180057@st.usst.edu.cn (T.T.); slzhuang@usst.edu.cn (S.Z.)
2 Department of Printing and Packaging Engineering, Shanghai Publishing and Printing College, Shanghai 200093, China; baichunyan@sppc.edu.cn
3 School of Artificial Intelligence and Manufacturing, Hechi University, No. 42 Longjiang Road, Yizhou District, Hechi 546300, China; sheng120410@163.com
4 Engineering Research Center of Optical Instrument and System, The Ministry of Education, Shanghai 200093, China
* Correspondence: dwzhang@usst.edu.cn

**Abstract:** Optical skyrmion lattices play an important role in photonic system design and have potential applications in optical transmission and storage. In this study, we propose a novel metasurface approach to calculating the dependence of the multi-beam interference principle and the angular momentum action in the spin–orbit interaction. The metasurface consists of nanopore structures, which are used to generate an optical skyrmion lattice. The superposition of optical vortex beams with circular polarization states is used to evaluate the evolution of the shape of the topological domain walls of the hexagonal skyrmion lattice. Our results show that the distribution of the skyrmion spin vector can be controlled by changing the lattice arrangement from triangular to hexagonal shapes. The distribution of skyrmion number at the microscale is further calculated. Our work has significant implications for the regulation of the shape of topological domain walls of skyrmion lattices, with potential applications in polarization sensing, nanopositioning, and super-resolution microimaging.

**Keywords:** metasurface; optical skyrmion lattices; topological domain walls; skyrmion number





## 1. Introduction

Skyrmions are field distributions with no standard smooth shape and can be explained by topological defects in the exotic phases of matter [1], cosmology [2], and the optical field. Optical skyrmion [3,4], as a type of topological defect, is a topologically protected two-dimensional and three-component vector field. Since the theory of magnetic skyrmion [5–7] in Bose–Einstein condensates [8] and liquid crystals [9] was confirmed, they have been extensively studied in the field of optical information storage due to their flexibility and low current drive.

Lattice light fields, on the other hand, are special fields with periodic or periodic-type structures [10] that are widely used in various physics fields. Plasmon metasurfaces [11,12], as artificial materials developed with artificial intelligence, have the advantage of efficiently regulating the electromagnetic field wave-front. By controlling the rotation of the structural unit, the emergent light can be changed, including the phase, amplitude, and polarization state [13–15]. The combination of the Pancharatnam–Berry phase [16] and other phases in plasmon metasurfaces provides better flexibility in the light field regulation compared to reflective liquid-crystal spatial light modulators (LCSLMs). The generation of broadband achromatic lenses [17] and high numerical aperture objectives [18] using metasurface has been demonstrated.

The general method for the shape transformation of optical skyrmions is using polarization singularities and phases [19,20]. However, these methods have limitations in terms of flexibility in light fields' phase regulation.

## 2. Structure Design and Principle Analysis

In this study, we propose a method to generate optical skyrmion lattices by superimposing the orbital angular momentum (OAM) of the incident beams. The wave amplitudes and wavelengths of the incident beams are taken into consideration, and the j-order beam azimuth angle is calculated.

The wave amplitudes are assumed to be $A$; the wavelength is $\lambda$; the $j$-order beam azimuth angle is $\theta_j$; the topological charge is $l$; the initial phase is $l\theta_j$; the $x$-axis and $y$-axis direction unit vectors are, respectively, $\vec{x}$ and $\vec{y}$; the wave vector size is $k = 2\pi/\lambda$; the angular coordinate of the complex amplitude field of the interference field is $\alpha$. The position vector is $\vec{r} = r\left(cos\alpha\,\vec{x} + sin\alpha\,\vec{y}\right)$, the $j$-order beam wave vector is $\vec{k}_j = k\left(cos\theta_j\,\vec{x} + sin\theta_j\,\vec{y}\right)$, and the two-dimensional complex amplitude field generated with the $N$ plane wave interference is $U_N(r, \alpha)$. Thus, we can obtain the following result: $\vec{k}_j \cdot \vec{r} = -krcos(\alpha - \theta_j)$. The complex amplitude field of the interference field is described mathematically in polar coordinates [21]:

$$U(r, \alpha) = \sum_{j=0}^{N-1} A \exp\left[i\left(\vec{k}_j \cdot \vec{r} + l\theta_j\right)\right] \tag{1}$$

After the Bessel transformation, it is expressed as follows:

$$\exp\left[-ikrcos(\alpha - \theta_j)\right] = \sum_{n=-\infty}^{n=+\infty} (-i)^n J_n(kr) \exp\left[in(\alpha - \theta_j)\right] \tag{2}$$

Equation (1) can be further expressed as Equation (3):

$$U_N(r, \alpha) = \sum_{j=0}^{N-1} \sum_{n=-\infty}^{n=\infty} A \exp(il\theta_j)(-i)^n \exp\left[in(\alpha - \theta_j)\right] J_n(kr) \tag{3}$$

This shows that the nature of the ideal lattice field generated via multi-beam interference depends on the number of interference beams, $N$, and the initial phase difference of the adjacent azimuth beam, $\Delta l\theta_j$. Considering $N\theta_j = (2j + 1)$, Equation (3) can be expanded as follows:

$$U_N(r, \alpha) = \sum_{n=-\infty}^{n=+\infty} (-i)^n A exp(il\theta_j) J_N(kr) \left\{ \begin{array}{l} \exp\left[in(\alpha - \frac{\pi}{N})\right] + \exp\left[in(\alpha - \frac{3\pi}{N})\right] + \ldots \\ + \exp\left[in\left(\alpha - \frac{(2N-3)\pi}{N}\right)\right] \\ + \exp\left[in\left(\alpha - \frac{(2N-1)\pi}{N}\right)\right] \end{array} \right\} \tag{4}$$

We can infer that some items offset; thus, Equation (3) can be further expanded as $NA \sum_{\varphi=-\infty}^{\varphi=+\infty} \exp(iN\varphi\alpha) J_{N\varphi}(kr)$ because $N$ is limited, the degeneracy of $N$ is (0.5N + 1), ($n = N\varphi$), and the interference beam can be further reduced to the following equation:

$$\begin{aligned} U(r, \alpha) &= \sum_{j=0}^{N-1} \sum_{n=-\infty}^{n=\infty} A \exp(il\theta_j)(-i)^n \exp\left[in(\alpha - \theta_j)\right] J_n(kr) \\ &= NA \sum_{\psi=-\infty}^{\psi=\infty} e^{(iN(\psi+l)\alpha)} J_{N\psi}(kr) \end{aligned} \tag{5}$$

In the above equation, $J_{N\varphi}$ is $N\varphi$ order Bessel function of the first kind. The superposition of orbital angular momentum states carrying different topological charges across an infinite dimensional space is the nature of the multi-beam interference. As far as we know, this theory is used for the regulation of the morphological generation of skyrmion lattices for the first time. To design metasurface structures with the coaxial superposition of orbital angular momentum [22], we used circularly polarized light vertically incident on a metal surface with an array of rectangular pores. The transmission field beam carries different

topological charges, and horizontal overlap and interference occur on the other side of the metasurface near the focused field plane. By changing the metasurface pores' spin, we can regulate the transverse skyrmion lattice field. The centers of the nanorectangular pores are specifically distributed on concentric ring bands, and the transmission field of each pore is calculated. Figure 1 illustrates this set of orthogonal rectangular pores after the incident of circularly polarized light $\sqrt{2}\begin{bmatrix}1 & \sigma i\end{bmatrix}^T/2$. The incident spin component of each transmitted light field remains constant, the transformed spin component is reversed and carries a geometric phase factor $\exp(2i\sigma\phi_n)$, and the central positions of the two rectangular pores are specifically distributed on the concentric ring bands. The transmission field E of two nanorectangular pores can be expressed using Equation (6):

$$\frac{1}{\sqrt{2}}\begin{bmatrix}1 & -\sigma i\end{bmatrix}^T \exp(2i\sigma\phi_n) = E_1 + \exp(i\pi)E_2 = E_{1,2} \tag{6}$$

where $E_1$ is the transmission field of the inner rectangular pores, and $E_2$ is the transmission field of the outer rectangular pores. The $\sigma$ parameter is related to the type of circularly polarized light; $\sigma$ is 1 when the circularly polarized light is left-handed, and $\sigma$ is $-1$ when the circularly polarized light is right-handed. The transmission phase change is introduced to ensure that the incident spin component of each transmitted light field remains constant. The geometric phase depends on the orientation angle size, and the inner and outer band nanorectangular pores maintain synchronous rotation to ensure the half-wave plate effect. The metasurface structure is carefully designed to achieve the desired transmission phase change. We arranged the centers of a set of nanorectangular pores on concentric rings along a radial distance, and the difference from the distance from the adjacent ring to the center of the focal field is half of the incident wavelength. The center of a group of nanorectangular pores is arranged in the same straight line as the center of the concentric circle; this particular rotational symmetric structure does not affect the half-wave plate effect. The design scheme of the specific structure of the metasurface is further described in the simulations presented in subsequent sections.

Here, we define the rotation order of the rectangular pores: $\Theta$ is the rotational order possessed with the different ring bands, which is noted as $\Theta_n$ (n = 1, 2, 3, . . .). In Figure 1b, $\phi$ is the orientation of the inner-ring nanorectangular pore, which is defined by the normal and horizontal axes of the long edges; the inner-ring orientation is rotated 90° clockwise relative to the orientation of the outer ring and thus generates the Pancharatnam–Berry geometric phase. We assume that $\phi_n = \Theta_n\theta + \phi_0$ (n = 1, 2, 3, ...); here, $\phi_n$ is the orientation angle, which directly determines the phase factor carried with the local linear polarization-converted spin component $\exp(2i\sigma\phi_n)$.

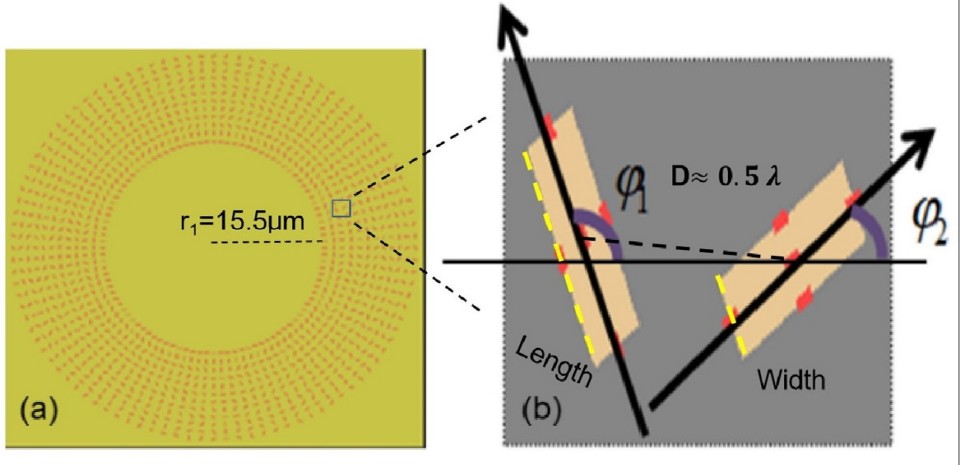

**Figure 1.** *Cont.*

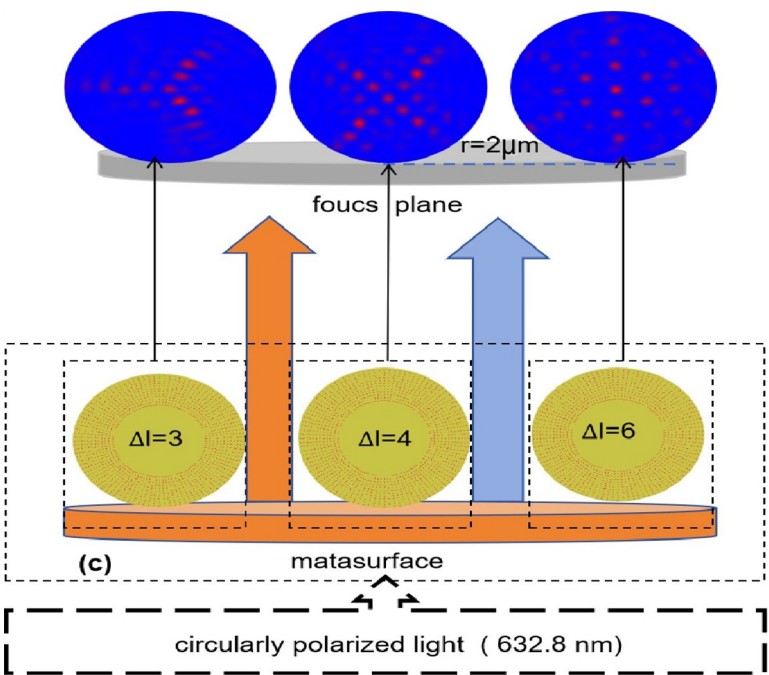

**Figure 1.** (**a**) Metasurface sample, the radius of the initial ring:$r_1$ is 15.5 μm; an array with 12 circles of rectangular pores; (**b**) schematic of nanopore structure: each pore with a length of 250 nm and a width of 90 nm, the wavelength is 632.8 nm, D is the near-half wavelength, and $\phi_n$ is the orientation angle according to the previous derivation for individual rectangular pores; (**c**) schematic of the regulation of topological domain walls of optical skyrmion lattices.

The array of metasurface rectangular pores is distributed on concentric rings, and each single ring band has 100 groups of rectangular pores distributed on the band. The rotation order of different ring bands is different, and the rotation order of the same ring band is constant, so the regulation of multi-beam ipsilateral axial OAM superposition of metasurface is realized to control the generated skyrmion lattice field. The diffraction field and the metasurface plane are represented in polar coordinates, and the diffraction focal length is assumed. According to the Huygens–Fresnel principle [23], the distribution of diffraction field electric field is $E(R, \alpha)$. The distribution of the diffractive electric field after circularly polarized light through the metasurface is equal to the superposition of the diffractive electric field distribution of transparent rectangular pores' array in each band. The diffraction field after the superposition of each ring band is focused at a certain distance behind the metasurface. The distribution of the diffraction field electric field was calculated, and the transmission field data were collected for further analysis.

The plane structure distribution of the metasurface is also shown with the polar coordinates $E(r, \theta)$. The diffraction focal length is assumed; $\lambda$ is the sum of the diffraction fields for each ring band and is equal to the total diffraction as follows:

$$E(R, \alpha) = \sum_{n=1}^{N} E_n(R, \alpha) = \frac{1}{j\lambda} \sum_{n=1}^{N} \int_0^{2\pi} E_n(R, \alpha) \frac{1}{l_n} K(\theta) \exp(ikl_n) r_n dr d\theta \qquad (7)$$

The distance size of the observed plane to the nanorectangular pores is two orders of magnitude of the aperture size, and the range of the tension angle of any point in the geometric range of a single nanopore is small under the paraxial approximation, $K(\theta) \approx 1$. According to Figure 1, the distance from the center position to the focus position and

the size of our observation plane is negligible compared to the optical path; this result is approximated into Equation (8):

$$l_n = \left(R^2 + (l_n)^2 - 2Rl_n \cos \Psi\right)^{1/2}, \cos \Phi = \frac{\vec{R} \cdot \left(\vec{l}_n\right)}{\left|\vec{R}\right| \cdot \left|\vec{l}_n\right|} \tag{8}$$

and thus the simplified version is obtained, which is expressed in Equation (9):

$$
\begin{aligned}
E(R, \alpha) &= \frac{1}{j\lambda} \sum_{n=1}^{N} \int_0^{2\pi} E_n(R, \alpha) \frac{1}{l_n} K(\theta) \exp\left[ ik\left(R^2 + (l_n)^2 - 2Rr_n \cos(\alpha - \theta)\right)^{\frac{1}{2}}\right] r_n dr d\theta \\
&= \frac{1}{j\lambda\tilde{\chi}} \sum_{n=1}^{N} \int_0^{2\pi} E_n(R, \alpha) \exp\left[ ik\left(l_n - 2R\frac{r_n}{l_n} \cos(\alpha - \theta)\right)\right] r_n dr d\theta
\end{aligned} \tag{9}
$$

We can further derive Equation (10) from Equation (9):

$$E(R, \alpha) = \frac{1}{j\lambda\tilde{\chi}} \sum_{n=1}^{N} \exp(il_n) 2\pi i^{2\sigma\Theta_n} J_{2\sigma\Theta_n}\left(\sigma kR\frac{r_n}{l_n}\right) \exp(2i\sigma\Theta_n\theta)\mu^{-\sigma} \tag{10}$$

From the final derivation results, each ring band carries a different topological charge $2\sigma\Theta_n$, the multi-superposition of topological states with different topological charges occurs, and the value of the multi-superposition of topological states can control the morphological evolution of the topological lattice field.

## 3. Simulation and Discussions

To verify the methods presented above, numerical simulations were conducted, which are presented in this section.

### 3.1. Simulation Setup

We designed a metasurface structure and performed simulations using FDTD software (FDTD Solutions 2016a). We first excited a 200 nm thick aurum film with a right-handed circularly polarized light source. We arranged an array with 12 circles of rectangular pores on the metasurface at a certain rotation order to improve the intensity of the transmission field and enhance the morphological characteristics of the optical lattice field. Other parameters of the FDTD scheme were as follows: each pore on the aurum film had a length of 250 nm and a width of 90 nm; the wavelength was 632.8 nm; and perfect boundary conditions (PML) were applied in all directions with two-mesh accuracy. We set the 2D Z-normal frequency domain power monitor to collect the transmission field data at a distance of 12.2 μm from the bottom of the metasurface. The simulations were recorded separately as $S_3, S_4, S_5, S_6$. The order of the single-ring band rotation for each sample was set, respectively, as $\Theta_3, \Theta_4, \Theta_5, \Theta_6$; the distribution of the localized surface of each set of nanorectangular pores was carefully designed considering the transmission phase [24] $\pi$, each sample, and 100 rectangular pores, which were arranged according to the rotation order. The number of rings was odd and the number of corresponding rings was even, and the radius of the initial ring $r_1$ was 15.5 μm. In addition, the radius size of the other 11 single rings needs to obey the transmission phase $\pi$; thus, the radius recurrence relation is as follows:

$$r_n = \sqrt{r_1^2 + \frac{\lambda^2(n-1)^2}{4} + \sqrt{r_1^2 + \tilde{\chi}^2}\lambda(n-1)} \tag{11}$$

$$r_{n+1} = \sqrt{r_n^2 + \lambda\sqrt{r_n^2 + \tilde{\chi}^2} + \frac{\lambda^2}{4}} \tag{12}$$

where n represents a positive odd number of not more than 12, and $\tilde{\chi}$ is 12.1 μm. To achieve the regulation of topological states, the orbital angular momentum of the four samples was superimposed. According to the radius formula derived from the geometric relationship, the radius of 12 rings has a strict recursive relationship, and once the initial radius is

changed, not only will the radius of the other 11 rings change, but the distance between the radius of the two adjacent rings also changes. Obviously, the recursive relationship is still satisfied mathematically, but the radius of adjacent rings changes significantly, and the transmission phase [24] $\pi$ cannot be strictly maintained, which does not meet the requirements of our design. We initially set the initial radius to 10 μm, but we could not obtain the skyrmion lattice we wanted. Then, we tried setting it at 15 μm and 5 μm; fortunately, we obtained an approximate result at 15 μm. The effect was most significant under the initial radius setting of 15.2 μm to 15.6 μm. In the actual process of processing samples, there were certain problems. The date of the initial radius setting (15.5 μm) can be used to avoid exceeding the testing range due to processing problems.

We obtained the transformation of different forms of topological domain walls, including a single skyrmion form ($|l|_{min} = 0$). We observed the electric field intensity in the transmission focal field and obtained different forms of topological domain walls, including triangular, quadrangle, pentagonal, and hexagonal optical skyrmion lattice patterns.

### 3.2. Dynamical Regulation of the Shape of Topological Domain Walls of Skyrmion Lattice

We observed the image of the x-y plane transmission focused electric field intensity of each group of samples within a square range of 2 μm. From Figure 2c, we can infer that each single lattice is a standard hexagonal skyrmion pattern and a hexagonal array composed of seven bright spots, and each bright spot is a skyrmion of a lattice cell structure. Interestingly, we obtained topological lattices of various shapes, albeit from a macroscopic perspective.

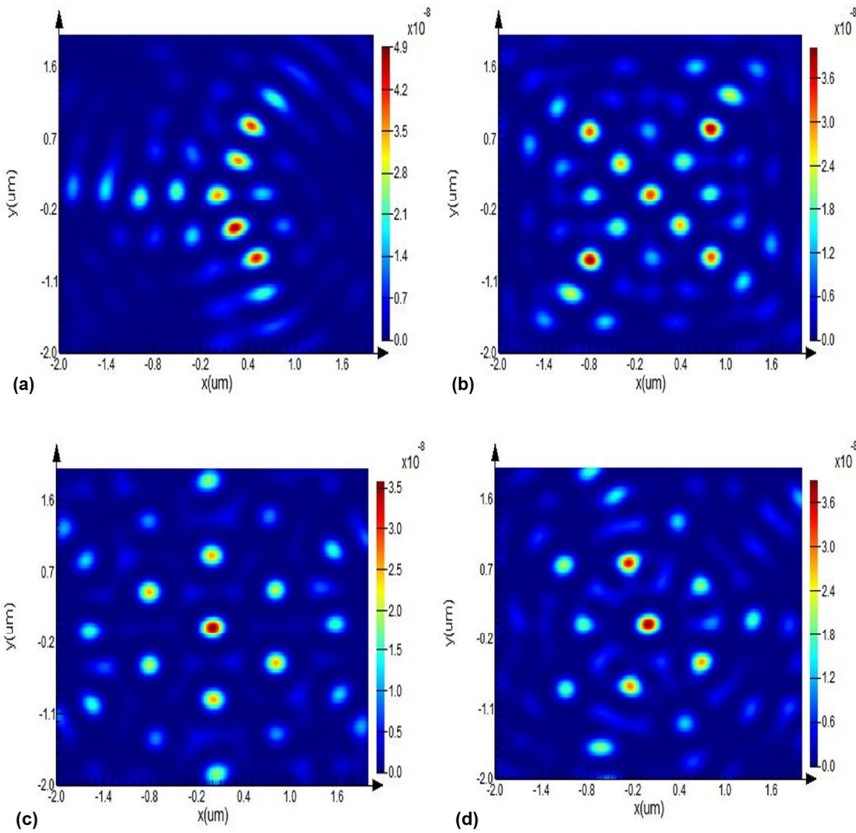

**Figure 2.** Electric field intensity diagram of the vector light field under spin-orbit action from the metasurface in the x-y plane: (**a**) ($|\Delta l| = 3$) triangular topological domain walls' optical skyrmion lattice pattern; (**b**) ($|\Delta l| = 4$) quadrangle topological domain walls' optical skyrmion lattice pattern; (**c**) ($|\Delta l| = 6$) hexagonal topological domain walls' optical skyrmion lattice pattern; (**d**) ($|\Delta l| = 5$) pentagonal topological domain walls' optical skyrmion lattice pattern.

### 3.3. Fine Structures of the Different Optical Skyrmion Lattice Patterns

However, to provide additional strong evidence that the generation of this method is skyrmion, we further narrowed down the acquisition region of the simulated field structure to obtain the spin vector distribution of the individual skyrmion. The set of orbital angular momentum was also satisfied $|\Delta l| = 6$, and other designs were controlled separately as $|\Delta l| = 3, 4, 5$, from different orbital angular momentum superposition states generated under coaxial; the fine structure had an obvious turn of vector direction between the generated skyrmion and the adjacent skyrmion structure, which was previously reported as the skyrmion topological charge fission that generated the topological domain walls.

For the first time, by controlling the orbital angular momentum superposition state, we were able to control the gradual transition of skyrmion topological domain walls from a triangle to a hexagonal type. Additionally, in the process, $|l|\min = 0$ is guaranteed; therefore, the topological domain walls do not change during continuous tuning, that is, the chiral changes and the topological domain walls morphology of skyrmion on the local surface are not correlated. This feature is of great significance for controlling the shape of the skyrmion lattice and the chirality of the skyrmion, making the two regulated with the discrete difference of the orbital angular momentum states and the peak of the orbital angular momentum. Figure 3 shows the fine structure of the skyrmion spin vector distribution in the square area.

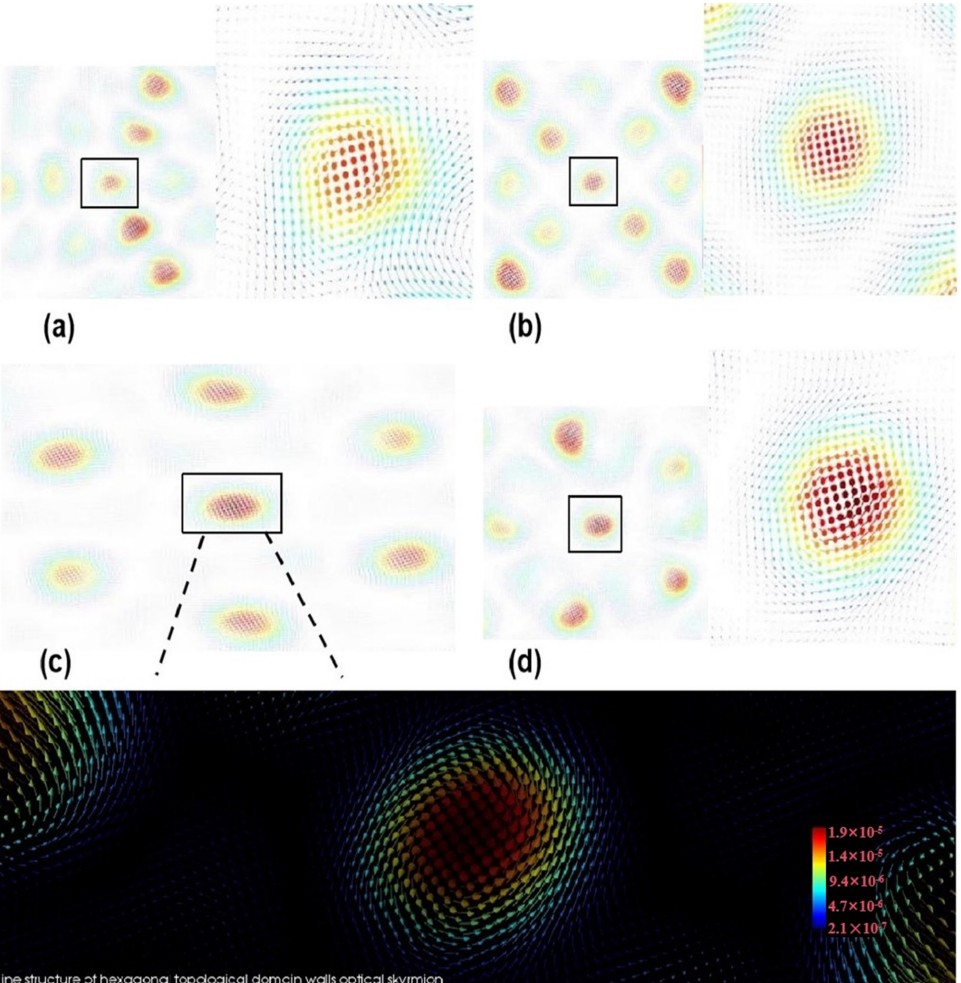

**Figure 3.** Fine structures of the different optical skyrmion lattice patterns: (**a**) the distribution of the spin vector of the triangular optical skyrmion; (**b**) the distribution of the spin vector of the quadrangle optical skyrmion; (**c**) the distribution of the spin vector of the hexagonal optical skyrmion; (**d**) the distribution of the spin vector of the pentagonal optical skyrmion.

We also observed that the fine structure of the skyrmion was perfectly bound within the polygon boundary. The optical skyrmion field vectors are divided into two types in a solid-state system: The Néel skyrmion provides cycloidal vector rotation, and the Bloch skyrmion provides angular vector rotation. Due to the existence of boundary and symmetry conditions, different skyrmion lattices show different field-vector helical reversal characteristics, and the anti-skyrmion orientation is observed in tetragonal Heusler materials [25]. In this case, the cylindrical symmetry is broken, and the inverse skyrmion field exhibits the combinatorial behavior of the cycloid and helical vectors. It is clear that the vector helix reversal in Figure 3 is in the special direction, and the central position rotates along the axis cycloid, carrying both skyrmion features of Néel and Bloch types. The vector direction is obviously inverted at the topological domain walls' boundary, which is due to the topological charge fission [20].

### 3.4. Calculation of the Distribution of Skyrmion Number

To quantitatively investigate the different characteristic skyrmion lattices generated with this approach, we first used the acquired transmitted electric field as a source to calculate the skyrmion density [26]:

$$N_s \vec{r} = \frac{1}{4\pi} \vec{e} \cdot \left( \frac{\partial \vec{e}}{\partial x} \times \frac{\partial \vec{e}}{\partial y} \right) \tag{13}$$

Four sets of samples, all corresponding to six rings, were generated, with each loop explained using the half-wave plate of characteristic periodic arrangement.

According to the spin–orbit interaction, the calculated skyrmion number depends on the superposition of the spin–orbit interaction, although the intensity of the diffraction field is significantly lower than the surface plasmon interference field, which does not affect the topological stability of the hybrid skyrmion. The normalized point diffraction electric field Stokes vector [27] is as follows:

$$E = \sum_{i=1}^{6} \left[ \frac{E_{2i-1}{}^2 - E_{2i}{}^2}{E_{2i-1}{}^2 + E_{2i}{}^2} + \frac{2E_{2i-1}E_{2i}}{E_{2i-1}{}^2 + E_{2i}{}^2} \times \begin{pmatrix} -\sin 2\theta \sin 2\Delta l\theta \\ -\cos 2\theta \sin \Delta l\theta \cos \delta - \sin \Delta l\theta \sin \delta \\ -\cos 2\theta \sin \Delta l\theta \cos \delta + \sin \Delta l\theta \cos \delta \end{pmatrix} \right] \tag{14}$$

Within the two-dimensional plane, $E_i$ is the angular direction with a value ranging from 0 to $2\pi$, and radials are $r_n (n = 1, 2, \ldots, 12)$.

$$
\begin{aligned}
N_{sk} &= \frac{1}{4\pi} \iint_A e \cdot [\partial_x e \times \partial_y e] \, dx \, dy \\
&= \sum_{i=1}^{6} \frac{1}{4\pi} \iint_A 4\Delta l \frac{E_{2i-1}E_{2i}[E_{2i-1}\partial_r E_{2i-1} \partial_r E_{2i-1}E_{2i}]}{[E_{2i-1}{}^2 + E_{2i}{}^2]} \, dr \, d\theta \\
&= \sum_{i=1}^{6} \Delta l \left[ \frac{E_{2i-1}(r_{2i-1})^2}{E_{2i-1}(r_{2i-1})^2 + E_{2i}(r_{2i-1})^2} + \frac{E_{2i-1}(r_{2i})^2}{E_{2i-1}(r_{2i})^2 + E_{2i}(r_{2i})^2} \right]
\end{aligned}
\tag{15}
$$

From the spin vector pattern, it can be seen that the generated spin vector combines the characteristics of both Néel and Bloch types' transverse spin out of the plane and transverse spin in the plane. This type of skyrmion with two characteristics simultaneously at a two-dimensional position may appear unconvincing when calculating the number at the position, but it is not difficult to find the correlation between the calculation formula of skyrmion and angular momentum. As can be seen in the formula, the skyrmion number and OAM difference are proportional.

As shown in Figure 4, we used the data of electric field to calculate the distribution of skyrmion with normalization of the electric field intensity from different lattices and found that the arrangement of skyrmion in each single lattice was tight in the plane, especially within a range of angles. This regular arrangement caused other skyrmion lattices to interact with each other. In addition, the tight arrangement formed a gap band of skyrmion, shown in blue. These gap parts are related to the red parts. The blue parts represent the topological domain walls mentioned above, and the connected red parts of the skyrmion lattice are continuously regulated due to changes within the angular momentum action in

spin–orbit interaction. The distribution of numerical values is discrete and exhibits a clear trend of motion.

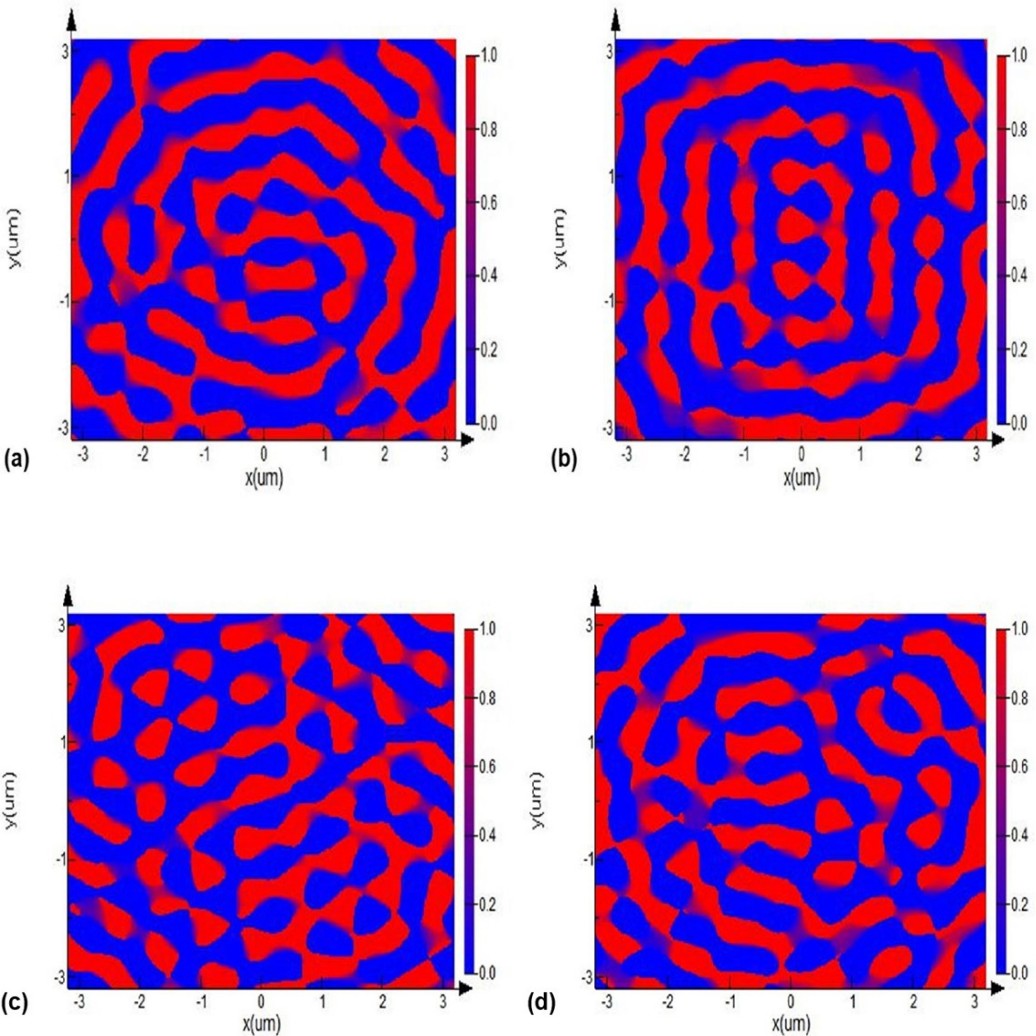

**Figure 4.** The calculated distribution of the skyrmion number with the normalization of the electric field intensity from different lattices: (**a**) the calculated distribution of the skyrmion number of tri angular optical skyrmion lattice; (**b**) the calculated distribution of the skyrmion number of quadrangle optical skyrmion lattice; (**c**) the calculated distribution of the skyrmion number of hexagonal optical skyrmion lattice; (**d**) the calculated distribution of the skyrmion number of pentagonal optical skyrmion lattice.

It can be clearly seen in Figure 4 that the distribution of skyrmion numbers in different lattice shapes is regulated with the angular direction, demonstrating the feasibility of using the design method considering the orbital angular momentum differences to obtain the distribution of skyrmions in different topological states. Under the triangle topological texture, the skyrmion at the space corner position, along with the changing shape of the skyrmion lattice, correspondingly, present other spatial angle distribution features; the hexagon optical skyrmion lattice shows six enriched regions, and the quadrangle optical skyrmion lattice shows four enriched regions. The probability of skyrmion occurrence outside of these enriched regions is low, and its geometric positional enrichment is evident in the three domains. These findings further illustrate the advantage of the proposed method for the generation of optical skyrmion fields, which relies on the orbital angular momentum; it can complete the topological form's continuous tuning and proves the regularity of optical skyrmion number distribution. Due to the stability of the chirality and

the topological protection, our research has significance in the modern optical information communication field.

## 4. Conclusions

In conclusion, we proposed a novel method to dynamically control the optical skyrmion lattice by superimposing the orbital angular momentum of the incident beams. It is interesting to note that this method is very similar to the interference of multiple beams, and it can perfectly produce skyrmion lattices. Our method allows for the continuous tuning of the topological domain walls and the shape of the skyrmion lattice.

From the collected data regarding the electric field vector distribution, it was found that, in the process of the continuous tuning of optical skyrmion, the chirality of the generated skyrmion had both the characteristics of the Néel skyrmion cycloid vector rotation and the characteristics of the Bloch skyrmion angular vector rotation perfectly. We demonstrated the effectiveness of our method through simulations and calculations of the skyrmion number. Our method provides a new way of thinking in terms of the control of optical skyrmion lattices. In the next step, we expect to improve the characteristics of metasurfaces to improve the electric field intensity and reduce the background noise. Furthermore, we hope to perform the visualization of higher-order skyrmion forms. Our work has significant implications for the control and regulation of skyrmions in the two-dimensional plane, with potential applications in polarization sensing [28], nanopositioning [29], super-resolution microimaging, and optical storage [30].

**Supplementary Materials:** The following are available online at https://www.mdpi.com/article/10.3390/photonics10111259/s1.

**Author Contributions:** Conceptualization, C.B.; methodology, T.T.; software, validation, data curation, analysis, and writing—original draft preparation, G.T.; writing—review and editing, J.P.; supervision, S.Z. and D.Z. All authors have read and agreed to the published version of the manuscript.

**Funding:** This project was supported by the National Natural Science Foundation of China No. 62275160.

**Institutional Review Board Statement:** Not applicable.

**Informed Consent Statement:** Not applicable.

**Data Availability Statement:** Data are contained within the article. Details see Supplementary Materials.

**Acknowledgments:** The authors would like to thank the University of Shanghai for Science and Technology for helping to identify collaborators for this work. All individuals included in this section have consented to the acknowledgements.

**Conflicts of Interest:** The authors declare no conflict of interest.

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
