# Peer review of "A Simulation Study of the Dynamical Control of Optical Skyrmion Lattices through the Superposition of Optical Vortex Beams"

_photonics, doi:10.3390/photonics10111259_

Round 1

Reviewer 1 Report

Comments and Suggestions for Authors

The manuscript “Dynamical Control of Optical Skyrmion Lattices through Su-2 perposition of the Optical Vortex Beams “ by G. Tang et al. concerns the study skyrmionics states on metasurface using optical vortex beams with circular polarization states. Authors have proposed a method to generate and dynamically control the optical skyrmion lattice by superimposing the orbital angular momentum of the incident beams, which allowed tuning of topological domain walls and the shape of skyrmion lattice. The presented research is timely, skyrmions are of considerable interest due to the possibility of their application in high – tech spintronic technologies.  The manuscript is good and reasoned written, authors obtained original and interesting results, the subject is suitable for Photonics journal, and I recommend the manuscript for publication.

Several remarks

1.               It should be added that incorporating some experimental results to validate the feasibility of the theory.

2.               It is not entirely clear from the text of the article for which materials this method can be implemented. Is crystal symmetry taken into account when studying different types of skyrmion lattices?

3.               There are several misprints in the article that should be corrected (“Daynamical regulation”,…)

Reviewer 2 Report

Comments and Suggestions for Authors

Optical skyrmion lattices play an important role in photonic system design and have potential applications in optical transmission and storage. In this study, we propose a novel metasurface approach to calculating the dependence of the multi-beam interference principle and the spin orbital angular momentum action. The metasurface consists of nanopore structures, which are used to generate an optical skyrmion lattice. The superposition of optical vortex beams with circular polarization states is used to evaluate the evolution of the shape of the topological domain walls of hexagonal skyrmion lattice. Results show that the distribution of skyrmion spin vector can be controlled by changing the lattice arrangement from triangular to hexagonal shapes. The distribution of skyrmion number at the microscale is further calculated. This work is interesting and in whole well organized, here below are some minor comments:

(1) The geometric details can be added.

(2) The simulation setup can be supplemented.

(3) It would be better for the authors to add the calculation method for the skyrmion number.

(4) Metasurfaces might be also a feasible way to obtain effective skyrmions, and the related works might be useful, for example DOI: 10.1002/lpor.202300152, and DOI: 10.1186/s43593-022-00013-3

Reviewer 3 Report

Comments and Suggestions for Authors

In the paper "Dynamical Control of Optical Skyrmion Lattices through Superposition of the Optical Vortex Beams" G. Tang et al. numerically simulated the metasurface designed for creation of skyrmion lattice. The paper could be of interest for researchers in the field of singular optics, but does not contain some important information and could not be published in its current form. The following major comments should be taken into account:

1. The transition from Eq. (1) to Eq. (3) should be explained in details

2. How this structure could be manufactured? Is it real to produce a structure with a size of 15.5 nm?

There are also some minor mistakes:

1. Different font size in Eqs. and main text

2. Parameters of FDTD scheme should be added to the manuscript

3. "microns" should be changed to "um"

4. There are some typos in the manuscript.

5. It is necessary to add a description of all quantities: l in Eq.(1), sigma in Eq. (4), etc.

Round 2

Reviewer 3 Report

Comments and Suggestions for Authors

The paper could be published